# Interpretable predictions from whole-body FDG-PET/CT using parameters associated with clinical outcome

Sambit Tarai[1] ✉, Elin Lundström[1], Nouman Ahmad[1], Robin Strand[2], Håkan Ahlström[1,3] & Joel Kullberg[1,3,4]

## Abstract

**Background** Accurate prediction of clinical outcomes is challenging yet important for patient care. The aim of the study was to evaluate a deep learning-based methodology using tissue-wise information, as a proof of concept, for predicting parameters known to be associated with clinical outcomes.

**Methods** We utilized the publicly available autoPET cohort, consisting of 1014 FDG-PET/CT examinations. Tissue-wise projections were extracted, representing specific tissues (bone, lean tissue, adipose tissue, and air) at different angles. A deep regression and classification framework was trained to predict total metabolic tumor volume (TMTV), lesion count, patient age, sex, and diagnosis status (cancer vs. no cancer). Saliency analysis was performed to identify image regions contributing most to each prediction.

**Results** Here we show that the best model predicts TMTV (MAE = 77 ml; $R^2$ = 0.84; (p <0.05)) and lesion count (MAE = 5.18, $R^2$ = 0.90), when using all tissue-wise projections. Age prediction improves when multiple projection angles are included (MAE = 6.57 years; $R^2$ = 0.70 (p <0.05)). The model also predicts sex (AUC = 1.00 (p <0.05)) and diagnosis status with high accuracy (AUC = 0.95 (p <0.05)).

**Conclusions** This proof-of-concept study demonstrates that tissue-wise projections can be used for efficient and automated prediction of parameters related to clinical outcomes, highlighting their potential for future prediction of clinical outcomes.

## Plain language summary

Cancer is a leading global cause of death. Physicians use images obtained from medical scans to detect cancer, guide treatment, and monitor outcomes. Artificial intelligence (AI) can support this process, but its decision making is often difficult to interpret. We have developed an AI model that first converts 3D scans into 2D projections. The AI model then utilizes these projections to predict key health parameters related to outcome, such as tumor volume, age, sex and cancer status. Our results demonstrate that the AI is accurate while also providing the ability to visualize the specific image regions it identifies as important for each prediction task. This transparency enhances trust in AI and demonstrates its potential to support personalized cancer care.

Clinical outcome prediction is a complex task that involves the utilization of diverse clinical data and medical imaging to forsee the probable outcome of a disease. Medical imaging contains abundant information that could indicate future outcomes. Positron emission tomography and computed tomography (PET/CT) with injected [18]F-fluorodeoxyglucose (FDG) is a combined imaging technique for cancer diagnostics, staging, and longitudinal monitoring[1,2]. The morphological and functional information in whole-body FDG-PET/CT images can provide insights into future tumor growth and prognostic outcomes that may not be fully apparent to clinicians by visual inspection alone[3,4]. However, the comprehensive analysis of these large-scale 3D datasets in search of prognostic biomarkers is not practically feasible through manual inspection, creating a need for automated, data-driven methods.

Deep learning techniques have emerged as powerful tools for discovering latent associations between imaging data and clinical outcomes, such as overall survival, disease progression, and therapy response[5]. Convolutional neural networks (CNN) have shown great potential in predicting clinical parameters of interest. For instance, Langer et al.[6] employed a CNN to predict the biological age from whole-body MRI, investigating age-related anatomical changes. Ferrández et al.[7] used a similar framework to predict the probability of time to progression within 2 years from PET/CT scans. Diamant et al.[8] predicted treatment outcomes for head and neck cancer patients. A common strategy in these studies has been the use of 2D projections from 3D volumes as efficient network inputs. None of them has investigated the possibility of extracting supplementary information through complementary tissue-wise multi-channel projections, specifically aiming to enhance overall prediction quality.

[1]Radiology, Department of Surgical Sciences, Uppsala University, Uppsala, Sweden. [2]Department of Information Technology, Uppsala University, Uppsala, Sweden. [3]Antaros Medical AB, Mölndal, Sweden. [4]Department of Surgical Sciences, SciLifeLab, Uppsala University, Uppsala, Sweden. ✉e-mail: sambit.tarai@uu.se

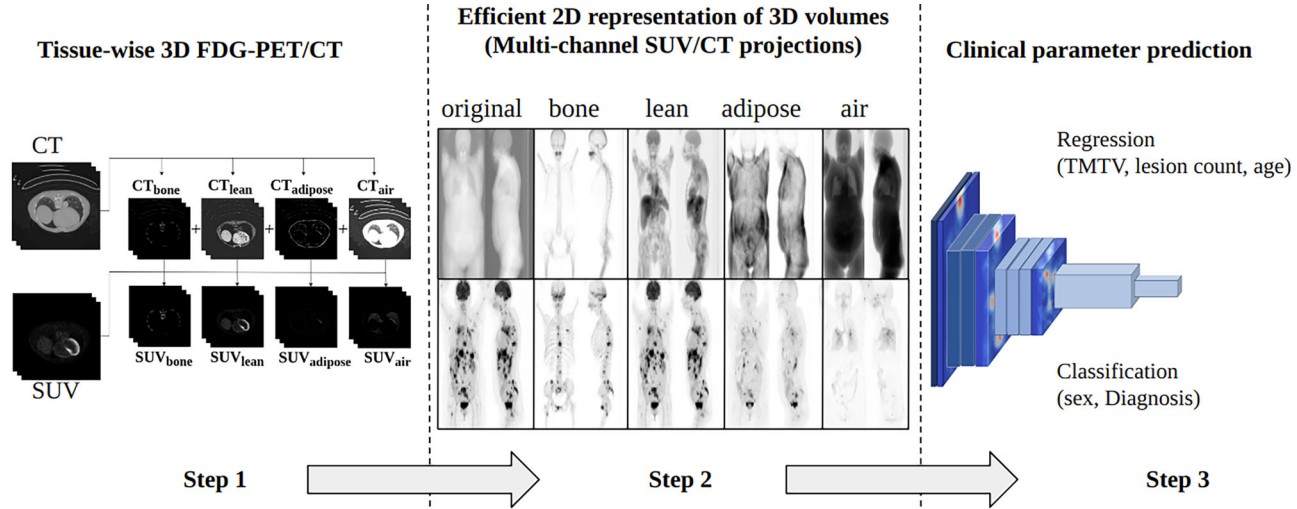

**Fig. 1 | Overview of the end-to-end automated framework for clinical parameter prediction from tissue-wise FDG-PET/CT projections.**

The autoPET[9] dataset (1014 oncologic FDG-PET/CT) offers a unique opportunity to explore the prediction of clinical outcome-related parameters from imaging data. Total metabolic tumor volume (TMTV) is one such parameter and a crucial biomarker in oncology, aiding in staging, prognostics and treatment planning[10,11]. Previous research has shown strong association of TMTV and lesion count with various clinical outcomes, including overall survival (OS)[12–18]. Automating the assessment of TMTV, using a deep regression framework, suggests its potential to also identify correlations between imaging data and crucial clinical outcomes, not readily available from the images or not apparent to human interpreters.

Studies have shown that age and sex are frequently incorporated as covariates in Cox proportional hazards models for survival analysis[19–21]. Unlike TMTV, which is visibly present in the image, age prediction is relatively challenging as it is not directly derivable from the images. If the network successfully can predict age, this indicates that it has learned associations that may not be obvious to humans. For instance, the discrepancy between predicted age and chronological age could offer information regarding the presence and severity of the disease. While the ability to predict TMTV and age demonstrates the capability of the deep regression framework, it also highlights the network's potential to capture meaningful associations between imaging data and clinical outcomes. Similarly classifying patients based on their sex and diagnosis status (disease vs non-disease) is also crucial. It not only serves as a foundation but also provides the opportunity for more complex categorizations in the future, including positive vs negative treatment outcomes. A preliminary study by our research group focused on explainable prediction of TMTV from FDG-PET/CT images[22]. In this study, we build on this work by predicting several parameters linked to clinical outcomes such as TMTV, lesion count, age, sex, and diagnosis status, to validate a proof-of-concept intended for future use in predicting clinical outcomes.

In this proof-of-concept study, we show that tissue-wise FDG-PET/CT projections could enable prediction of several clinically relevant parameters, including TMTV, lesion count, age, sex, and diagnosis status. Our saliency analysis further demonstrates that the model focuses on anatomically and clinically plausible regions, supporting the interpretability of the approach. Together, these findings holds promise for future applications in clinical outcome prediction.

## Methods

In this study, we explored the prediction of parameters associated with clinical outcomes, including TMTV, lesion count, age, sex, and diagnosis status, from whole-body FDG-PET/CT using a deep regression approach, as shown in Figure 1.

## Dataset

**Data collection.** This study utilized the publicly available autoPET dataset from The Cancer Imaging Archive (TCIA)[9,23,24]. Released in 2022 as part of a tumor segmentation grand challenge, the autoPET dataset consisted of 1014 FDG-PET/CT volumes, each accompanied by manually annotated tumor delineations. Alongside the segmentation masks, clinical parameters, such as age (in years), diagnosis type, and sex were also made available for each patient. The data encompassed three cancer types: lymphoma (144), lung cancer (167), melanoma (188), and a negative control group (513). All FDG-PET/CT and their manual annotations were provided as 3D volumes, typically from the head to the mid-thigh, and in some cases, the entire body, based on clinical relevance. The manual annotations were conducted by two radiologists with ten and five years of experience. Publication of the anonymized autoPET dataset was approved by the Institutional Ethics Committee of the Medical Faculty at the Eberhard Karls University of Tübingen and the University Hospital of Tübingen (833/2020BO2). All subjects signed an informed-consent form.

**Data acquisition.** PET/CT examinations were conducted at the University Hospital Tübingen using a Siemens Biograph mCT scanner[9]. The protocol adhered to international guidelines for oncologic PET/CT, specifically the FDG PET/CT EANM procedure guidelines version 2.0[25]. Diagnostic whole-body CT scans were acquired with standardized parameters: 200 mAs with automated exposure control (CareDose), tube voltage of 120 kV, and weight-adapted intravenous CT contrast agent (Ultravist 370, Bayer Healthcare) or without contrast agent (in case where contraindications existed). CT data were reconstructed in transverse orientation with a slice thickness ranging from 2.0 to 3.0 mm and an in-plane voxel size between 0.7 and 1.0 mm.

For PET imaging, FDG was intravenously administered after at least 6 hours of fasting, with a mean radioactivity of 314.7 MBq (range: 150 to 432 MBq) adjusted based on patient weight. PET scans covered four to eight bed positions (typically from skull base to mid-thigh), reconstructed using a 3D-ordered subset expectation maximization algorithm (2 iterations, 21 subsets, Gaussian filter 2.0 mm, matrix size 400 × 400, slice thickness 3.0 mm, voxel size 2.04 × 2.04 × 3 mm³). PET acquisition time was 2 min per bed position.

**Ethical considerations.** Ethical approval to conduct retrospective image analysis on the autoPET dataset was obtained from the Swedish Ethical Review Authority (Dnr 2023-02312-02). The study was conducted in accordance with relevant guidelines and regulations, including the Declaration of Helsinki.

## Data pre-processing

The PET data was standardized through the conversion to standardized uptake value (SUV), normalized by body weight. To ensure consistency with respect to voxel size throughout the entire cohort, all CT and SUV data were resampled to a common image resolution of $(2.04 \times 2.04 \times 3 \text{ mm}^3)$.

## Proposed method: Efficient 2D representation of 3D PET/CT scans

Due to the significant GPU memory requirements associated with processing 3D PET/CT volumes, 2D projections were derived. This resulted in suitable 2D representations of the underlying 3D volumes while minimizing overall memory usage. While 2D projections effectively can retain essential anatomical and metabolic information from the original PET/CT scans, it is important to acknowledge that complete data preservation cannot be obtained. To address this issue, supplementary information to the original CT ($CT_{orig}$) and SUV ($SUV_{orig}$) volumes, in the form of multi-channel 2D projections across various tissues was created. This involved categorizing the $CT_{orig}$ and $SUV_{orig}$ volumes into four distinct tissue types based on the CT Hounsfield units (HU) according to the equations (1)-(4)[26,27].

$$CT_i(bone) = \begin{cases} 1 & \text{if } i \geq 200 \\ 0 & \text{elsewhere} \end{cases} \quad (1)$$

$$CT_i(lean) = \begin{cases} 1 & \text{if } i \in [-29, 150] \\ 0 & \text{elsewhere} \end{cases} \quad (2)$$

$$CT_i(adipose) = \begin{cases} 1 & \text{if } i \in [-190, -30] \\ 0 & \text{elsewhere} \end{cases} \quad (3)$$

$$CT_i(air) = \begin{cases} 1 & \text{if } i < -190 \\ 0 & \text{elsewhere} \end{cases} \quad (4)$$

This categorization resulted in a set of tissue-wise CT masks, representing bone tissue, lean soft tissue, adipose tissue, and air. Subsequently, tissue-wise CT and SUV volumes were generated by applying these masks to the respective $CT_{orig}$ and $SUV_{orig}$ volumes. Finally, maximum intensity projections (MIPs) were computed for all the SUV channels by capturing the highest intensity along the coronal ($\Theta = 0°$) and sagittal ($\Theta = 90°$) directions. Similarly, mean intensity projections (meanIP) were computed for all the CT channels by capturing the mean intensity along the respective direction. This process resulted in the creation of tissue-wise 2D projections, including $CT_{orig}^{meanIP}$, $SUV_{orig}^{MIP}$, $CT_{bone}^{meanIP}$, $SUV_{bone}^{MIP}$, $CT_{lean}^{meanIP}$, $SUV_{lean}^{MIP}$, $CT_{adipose}^{meanIP}$, $SUV_{adipose}^{MIP}$, $CT_{air}^{meanIP}$, and $SUV_{air}^{MIP}$.

The decision to include these specific tissue categories was based on the fact that they are important tissue types that can be easily identified from CT based on their HU. These categories also represent key anatomical components that provide valuable insights into structural and metabolic aspects of the whole-body and have the potential to aid future clinical outcome prediction. The selection of the specific projection angles, corresponding to coronal (0°) and sagittal (90°) views was based on their ability to provide complementary and comprehensive perspectives of the whole body PET/CT. These projections are utilized in medical imaging by experts, including radiologists, due to their effectiveness in visualizing various aspects of internal anatomy.

One of the important aspects of this research was to examine the effectiveness of incorporating multi-directional tissue-wise projections in the context of clinical parameter prediction. Multi-directional 2D projections refer to multiple CT or SUV projections obtained at different angles $\Theta$ with respect to the sagittal plane, where $\Theta$ ranges between $[-90°, +90°]$[28,29]. Besides examining pure sagittal and coronal projections, we also evaluated the impact of introducing oblique projections, with $\Theta = \pm 45°$. This idea originated from the fact that coronal and sagittal projections offer distinct and complementary information to each other. Thus, the integration of oblique projections could potentially yield additional information.

## Clinical parameter prediction: TMTV, lesion count, age, sex, diagnosis status

TMTV, lesion count and age prediction were formulated as regression tasks, given their nature as continuous variables. In contrast, prediction of sex and diagnosis status was approached as classification tasks. The number of images used for both the classification tasks was 1010.

**Regression.** TMTV and lesion count exhibit a positively skewed distribution, with values ranging from 0.1 to 2481 ml for TMTV and 1 to 900 for lesion count. In contrast, age shows an approximately Gaussian distribution, spanning from 11 to 95 years. The number of images used for age prediction was 1010 whereas TMTV and lesion count prediction employed 499 images.

**Classification.** Diagnosis status was approached as a binary classification task, distinguishing between cancer-positive and cancer-negative categories. Similarly, sex classification comprised male and female categories. The evaluation of diagnosis status involved 1014 scans, while sex classification utilized 1010 scans, excluding four cases with unspecified sex information.

## Neural network training

**Data preparation.** MIPs and meanIPs were generated from tissue-wise SUV and CT volumes as discussed in section 2.3 along coronal and sagittal directions and normalized between [0, 1]. Subsequently, all the multi-channel coronal and sagittal projections were combined into a single image by placing them next to each other in the form of a collage which resulted in a total of 1014 collages. This was done independently for all the CT and SUV projections, resulting in 10 channel image collages for each patient, as shown in step 2 of Fig. 1.

Given the substantial variation in the PET/CT field of view, often encompassing regions from the head to mid-thigh, and occasionally extending across the entire body, a simple padding based approach was implemented. This involved zero padding and unpadding to the image collages, ensuring a uniform dimension of (512, 512) across the cohort. Subsequently, to assess the effectiveness of the integration of multi-directional projections, oblique projections ($\Theta = \pm 45°$) were introduced into the collage. This resulted in a uniform dimension of (512, 1024) across the entire cohort.

**Network configuration.** A 2D CNN based architecture named DenseNet-121[30] was utilized for all the prediction tasks, including TMTV, lesion count, age, sex, and diagnosis status. In all the cases, the input to the network was a combination of the multi-channel 2D collages consisting of $CT_{orig}^{meanIP}$, $SUV_{orig}^{MIP}$, $CT_{bone}^{meanIP}$, $SUV_{bone}^{MIP}$, $CT_{lean}^{meanIP}$, $SUV_{lean}^{MIP}$, $CT_{adipose}^{meanIP}$, $SUV_{adipose}^{MIP}$, $CT_{air}^{meanIP}$, and $SUV_{air}^{MIP}$. To evaluate the effectiveness of tissue-wise projections, several networks were evaluated, with the same network architecture, except for the number of input channels. These ablation experiments were primarily conducted for TMTV and age prediction, to investigate the impact of each individual channel. In all the prediction tasks, the baseline model utilized either the $SUV_{orig}^{MIP}$ or $CT_{orig}^{MIP}$ as input, while the proposed model incorporated all tissue-wise CT and SUV projections.

To ensure consistency and unbiased evaluation, all models were independently assessed using 10-fold cross-validation, employing the same training-validation split. Stratification based on sex and cancer type was applied during cross-validation to maintain the same distribution across all the folds. All networks were trained on a Nvidia RTX 3090 Ti GPU with 24 GBs of memory. The training process included a smooth L1 loss for regression and binary cross-entropy loss for classification tasks. Both tasks were optimized using the Adam optimizer with a learning rate of 0.0001, weight decay of 0.00001 and dropout rate of 0.25. A batch size of 30 was employed during training for faster convergence.

**Evaluation metrics.** During 10-fold cross-validation, various evaluation metrics were employed to facilitate a comprehensive comparison between different models. For regression tasks, the metrics included mean absolute error (MAE), coefficient of determination ($R^2$), and Pearson's correlation coefficient (r), providing insights into predictive accuracy and consistency. For classification tasks, the metrics included area under the curve (AUC), sensitivity, precision, and specificity.

## Saliency analysis: Grad-CAM

To visualize the regions within the input images that contributed most to the neural network's predictions, saliency analysis was performed using the Gradient-weighted Class Activation Mapping (Grad-CAM) approach. This algorithm computes the importance of a chosen feature map in the CNN in relation to the target variable. In our case, this was achieved by calculating the gradients of the output with respect to the feature map of the first convolutional layer. It generates a heatmap, indicating the regions of the input image that positively influenced the target prediction.

Analyzing individual saliency maps can be time-consuming and sometimes challenging to interpret in large cohorts. Therefore, cohort saliency analysis was employed to enhance overall understanding of recurring patterns across the entire cohort[22,31]. Cohort saliency analysis involved spatially aligning all individual saliency maps into a reference image using a previously developed whole-body FDG-PET/CT image registration framework[26]. Cohort saliency analysis allows us to visualize the overall patterns in the entire cohort by aligning individual saliency maps to a common reference, highlighting recurring regions of interest that contribute to the model's predictions across multiple subjects. Females and males were registered to separate reference images due to anatomical differences, with template selection based on body fat percentage[32]. Cost function masking was applied around individual tumors to preserve their original shape. Further details regarding image registration and template selection for the autoPET cohort can be found in ref. 33.

The image registration technique was originally developed for 3D FDG-PET/CT data. However, in this study, where 2D projections were utilized for deep regression-based prediction, we first converted the 2D saliency maps into 3D, registered them in 3D, aggregated and then converted them back to 2D. This was made possible because nearly all the subjects had a similar field of view in terms of anatomical coverage, with only minor differences. Further details regarding cohort saliency analysis in the autoPET cohort can be found in ref. 22.

## Statistics and Reproducibility

Prediction performance was assessed using $R^2$ and MAE for regression tasks (TMTV and age), and AUC for classification tasks (sex and diagnosis status). Statistical comparisons between the baseline and proposed models were conducted using two-sided Steiger's Z-test for regression tasks and two-sided DeLong's test for classification tasks. A $p$-value less than 0.05 was considered statistically significant, and no adjustments for multiple comparisons were applied.

All experiments were performed on the autoPET dataset, consisting of 1014 whole-body FDG-PET/CT scans. Model evaluation was conducted using 10-fold cross-validation. Each fold was trained once using identical preprocessing, network architecture, and training procedures, and no additional repeated runs with different random seeds were performed. In this study, the 10 folds serve as independent experimental replicates, as each fold provides an independent evaluation of model performance.

## Results
### Regression

Table 1 presents the results from 10-fold cross-validation for TMTV prediction based on different combinations of tissue-wise projections from various angles. The findings showed that incorporating additional SUV channels enhanced TMTV prediction, with the highest performance achieved when all SUV-based projections were used. It also demonstrated that the $SUV_{lean}^{MIP}$ channel was the most influential after $SUV_{orig}^{MIP}$, followed by $SUV_{bone}^{MIP}$, $SUV_{air}^{MIP}$, and $SUV_{adipose}^{MIP}$ channels. Additionally, incorporating CT channels provided no advantage during TMTV prediction, as CT-based projections generally lacked the contrast needed to distinguish tumors from normal tissues. The table also includes TMTV predictions using 3D segmentation networks (3D UNet and nnUNet)[28,34,35] as benchmarks. Finally, it provides results from using the mean value of the entire cohort to estimate TMTV.

Table 2 presented the results from 10-fold cross-validation for lesion count prediction. It also included predictions using 3D segmentation networks (3D UNet and nnUNet) as benchmarks.

## Table 1 | Comparison of Total Metabolic Tumor Volume (TMTV) prediction results from 10-fold cross-validation using tissue-wise CT and SUV projections from various angles

| Methodology | Inputs ($CT^{meanIP}$/$SUV^{MIP}$) | | | | | MAE (ml) (95% CI) | $R^2$ (95% CI) | r (95% CI) |
|---|---|---|---|---|---|---|---|---|
| | original | bone | lean | adipose | air | | | |
| [1] Input is a collage of [0°, 90°] (Densenet121) | −/✓ | −/✓ | −/✓ | −/✓ | −/✓ | 77 (71–91) | 0.84 (0.77–0.88) | 0.92 (0.90–0.93) (*) |
| [2] Same as above | ✓/− | ✓/− | ✓/− | ✓/− | ✓/− | 225 (190–257) | 0.01 (0.00–0.03) | 0.18 (0.05–0.33) |
| [3] Same as above | −/✓ | −/− | −/− | −/− | −/− | 88 (74–103) | 0.71 (0.59–0.80) | 0.85 (0.82–0.87) |
| [4] Same as above | −/✓ | −/✓ | −/− | −/− | −/− | 89 (76–101) | 0.74 (0.63–0.82) | 0.87 (0.84–0.89) |
| [5] Same as above | −/✓ | −/− | −/✓ | −/− | −/− | 86 (74–97) | 0.76 (0.63–0.85) | 0.87 (0.85–0.89) |
| [6] Same as above | −/✓ | −/− | −/− | −/✓ | −/− | 92 (78–106) | 0.71 (0.60–0.80) | 0.86 (0.84–0.88) |
| [7] Same as above | −/✓ | −/− | −/− | −/− | −/✓ | 97 (76–112) | 0.69 (0.59–0.75) | 0.83 (0.81–0.86) |
| [8] Same as above | ✓/− | −/− | −/− | −/− | −/− | 238 (191–280) | 0 (−0.13–0.01) | 0.16 (0.02–0.29) |
| [9] Same as above | ✓/✓ | ✓/✓ | ✓/✓ | ✓/✓ | ✓/✓ | 79 (68–91) | 0.83 (0.74–0.87) | 0.92 (0.90–0.93) |
| [10] Input is [0°] projection only (Densenet121) | −/✓ | −/✓ | −/✓ | −/✓ | −/✓ | 80 (71–93) | 0.77 (0.72–0.84) | 0.88 (0.86–0.90) |
| [11] Input is a collage of [0°, − 45°, 45°, 90°] (Densenet121) | −/✓ | −/✓ | −/✓ | −/✓ | −/✓ | 78 (71–92) | 0.84 (0.78–0.89) | 0.91 (0.89–0.92) |
| [12] Segmentation network (3D UNET) | - | - | - | - | - | 63 (55–74) | 0.82 (0.75–0.88) | 0.92 (0.90–0.93) |
| [13] Segmentation network (nnUNET) | - | - | - | - | - | 55 (45–68) | 0.84 (0.79–0.89) | 0.93 (0.91–0.95) |
| [14] Guess the mean | - | - | - | - | - | 203.03 | −0.0075 | nan |

The table summarizes the Mean Absolute Error (MAE), coefficient of determination ($R^2$), and Pearson's Correlation Coefficient (r) for each methodology tested, along with their 95% confidence interval. The inputs include combinations of original, bone, lean, adipose, and air channels, evaluated using a DenseNet-121 model. The performance of 3D segmentation networks (3D UNet and nnUNet) is also included for benchmarking. Steiger's Z-test was conducted between model 1 (proposed) and 3 (baseline) where '*' indicated statistical significance ($p$-value < 0.05).

Figure 2 displayed the confusion matrix for TMTV prediction using the best-performing model from Table 1, which incorporated all the tissue-wise SUV projections. The matrix showed that most diagonal elements had high values, indicating accurate predictions where the predicted TMTV closely matched the ground truth.

The results from 10-fold cross-validation for age prediction, using different combinations of tissue-wise projections from various angles, are presented in Table 3. The findings indicated that incorporating tissue-wise CT and SUV projections improved age prediction, with the best performance achieved using all CT- and SUV-based projections.

Figure 3 displays the confusion matrix for age prediction using the best-performing model from Table 3, which incorporated all the tissue-wise CT and SUV projections. The matrix showed that most diagonal elements had high values, indicating accurate predictions where the predicted age closely matched the ground truth.

Figure 4 showed the scatterplots of ground truth versus predicted values for TMTV and age prediction. It demonstrated that the predicted values closely resembled the ground truth distributions, indicating accurate model performance.

### Classification

The results from 10-fold cross-validation for sex and diagnosis status classification, using tissue-wise projections, were presented in Table 4. The proposed model for sex classification achieved an AUC of 1, whereas for diagnosis status classification an AUC of 0.95 was achieved.

Figure 5 shows the Receiver operating curve (ROC) for sex and diagnosis status classification for one of the folds.

Supplementary Figs. 1–4 present example cases from the saliency analysis of the best-performing model across various tasks: TMTV prediction, age prediction, sex classification, and diagnosis status classification. The saliency maps highlighted the key regions that were most influential in the deep regression tasks.

Supplementary Figs. 5–7 present example cases from the saliency analysis where the model failed to accurately estimate the target variable across various tasks: TMTV prediction, age prediction, and diagnosis status classification. The saliency maps highlight the regions that misled the model, resulting in inaccurate predictions.

Figure 6 illustrated the cohort saliency maps along the coronal and sagittal directions for males and females across various tasks: [a] TMTV prediction, [b] age prediction, [c] sex classification, and [d] diagnosis status classification. In the case of TMTV prediction, the corresponding aggregated ground truth tumor segmentation projections are shown in Supplementary Fig. 8. The aggregated saliency maps for TMTV prediction qualitatively resembled the ground truth projections, implying that the model was effectively identifying the relevant tumor regions in the overall cohort that contributed to the TMTV.

### Discussion

Our study focused on utilizing a deep regression framework for the prediction of parameters associated with clinical outcomes from tissue-wise FDG-PET/CT. By integrating these multi-channel projections into a CNN, we observed significant improvements in the prediction accuracy of TMTV, lesion count, and age compared to using single-channel projections such as $SUV_{orig}^{MIP}$ and $CT_{orig}^{meanIP}$ alone. These single-channel projections, commonly

**Table 2 | Comparison of lesion count prediction results from 10-fold cross-validation using tissue-wise CT and SUV projections with 3D segmentation networks**

| Methodology | Inputs ($CT^{meanIP}$/$SUV^{MIP}$) | | | | | MAE (95% CI) | $R^2$ (95% CI) | r (95% CI) |
|---|---|---|---|---|---|---|---|---|
| | original | bone | lean | adipose | air | | | |
| Input is a collage of [0°, 90°] (Densenet121) | –/✓ | –/✓ | –/✓ | –/✓ | –/✓ | 5.18 (3.45–5.36) | 0.90 (0.81–0.95) | 0.89 (0.87–0.92) |
| Segmentation network (3D UNet) | - | - | - | - | - | 3.14 (3.04–4.18) | 0.83 (0.75–0.88) | 0.92 (0.90–0.93) |
| Segmentation network (nnUNet) | - | - | - | - | - | 3.05 (2.91–3.96) | 0.84 (0.77–0.90) | 0.93 (0.91–0.94) |

The table summarizes the Mean Absolute Error (MAE), coefficient of determination ($R^2$), and Pearson's Correlation Coefficient (r) for each methodology tested, along with their 95% confidence interval.

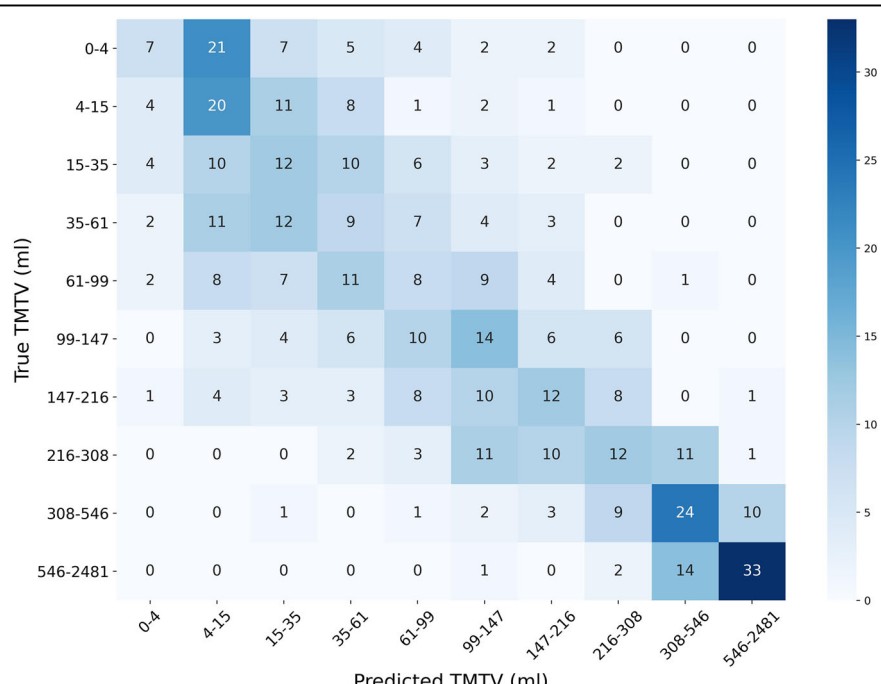

**Fig. 2 | Confusion matrix illustrating the prediction of Total Metabolic Tumor Volume (TMTV) using all tissue-wise SUV projections as input to a DenseNet-121 model.** The matrix displays the distribution of predicted TMTV values (in columns) against the ground truth values (in rows). Since, TMTV represents a continuous variable ranging from 0.1 to 2481 ml, it was grouped into several bins to facilitate the interpretation.

**Table 3 | Comparison of age prediction from 10-fold cross-validation using tissue-wise CT and SUV projections from various angles**

| Methodology | Inputs ($CT^{meanIP}$/$SUV^{MIP}$) | | | | | MAE (year) (95% CI) | $R^2$ (95% CI) | r (95% CI) |
|---|---|---|---|---|---|---|---|---|
| | original | bone | lean | adipose | air | | | |
| [1] Input is a collage of [0°, 90°] (Densenet121) | ✓/✓ | ✓/✓ | ✓/✓ | ✓/✓ | ✓/✓ | 6.57 (6.21–6.89) | 0.70 (0.65–0.73) | 0.84 (0.82–0.86) (*) |
| [2] Same as above | ✓/– | ✓/– | ✓/– | ✓/– | ✓/– | 7.35 (6.95–7.69) | 0.64 (0.59–0.68) | 0.80 (0.78–0.82) |
| [3] Same as above | –/✓ | –/– | –/– | –/– | –/– | 9.53 (9.07–10) | 0.42 (0.36–0.48) | 0.65 (0.62–0.69) |
| [4] Same as above | –/✓ | –/✓ | –/✓ | –/✓ | –/✓ | 8.25 (7.83–8.66) | 0.55 (0.49–0.60) | 0.75 (0.72–0.77) |
| [5] Same as above | ✓/– | –/– | –/– | –/– | –/– | 8.34 (7.91–8.74) | 0.54 (0.50–0.57) | 0.73 (0.71–0.76) |
| [6] Input is [0°] projection only (Densenet121) | ✓/✓ | ✓/✓ | ✓/✓ | ✓/✓ | ✓/✓ | 6.75 (6.36–7.14) | 0.67 (0.62–0.71) | 0.82 (0.80–0.84) |
| [7] Input is a collage of [0°, − 45°, 45°, 90°] projections (Densenet121) | ✓/✓ | ✓/✓ | ✓/✓ | ✓/✓ | ✓/✓ | 6.41 (6.03–6.75) | 0.72 (0.67–0.73) | 0.85 (0.83–0.86) |
| [8] Guess the mean | - | - | - | - | - | 44.64 | −0.0078 | nan |

It summarizes the Mean Absolute Error (MAE), coefficient of determination ($R^2$), and Pearson's Correlation Coefficient (r) for each methodology. The inputs include combinations of original, bone, lean, adipose, and air channels, evaluated using DenseNet-121 model. Steiger's Z-test was conducted between model 1 (proposed) and 5 (baseline) where '*' indicated statistical significance (p-value < 0.05).

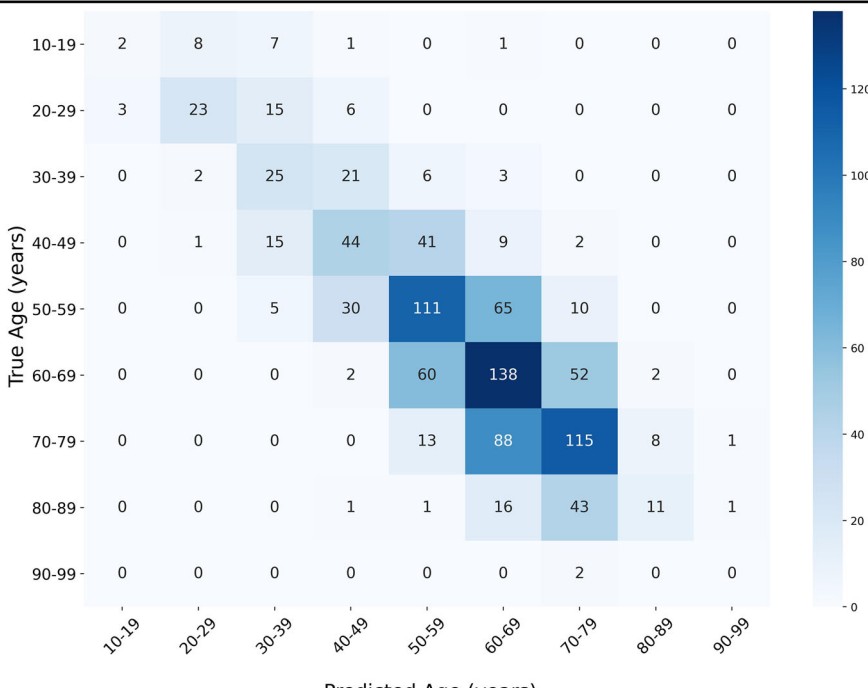

**Fig. 3 | Confusion matrix illustrating the prediction of age using all tissue-wise CT and SUV projections as input to a DenseNet-121 model.** The matrix displays the distribution of predicted age values (in columns) against the ground truth values (in rows). Since, age represents a continuous variable ranging from 11 to 95 years, it was grouped into several bins to facilitate the interpretation.

referenced in the literature, were outperformed by the tissue-wise multi-channel approach, as evidenced by the results in Tables 1, 2, and 3. The scatterplot illustration of the predicted values for TMTV and age closely resembles that of the ground truth data (see Fig. 4). Additionally, this approach proved effective for classification tasks related to sex and diagnosis status (see Table 4).

Automated TMTV prediction from whole-body FDG-PET/CT is important given its strong association with clinical outcomes in oncology[12–15,17]. It is a challenging task because of the limited spatial information present in the 2D projections, compared to 3D data, and considerable variation in TMTV. While the primary source of tumor-related information was the $SUV^{MIP}_{orig}$ channel, incorporating additional SUV projections improved the prediction of TMTV, increasing the $R^2$ from 0.71 to 0.84 and r from 0.85 to 0.92 (Steiger's test, p < 0.05), as shown in Table 1. This is because additional SUV channels offer metabolic insights of tumors across various tissues. They provide extra contrast to small and low FDG uptake lesions that are difficult to visualize in $SUV^{MIP}_{orig}$. This could be important for future outcome prediction tasks. Upon comparing our top-

performing model with the 3D segmentation networks 3D UNet (Dice = 0.70) and nnUNet (Dice = 0.74), we observed comparable performance (see Table 1)[34]. This proof-of-concept study demonstrates the strength of our 2D deep regression model, achieving comparable performance with less information. A detailed technical comparison between the 2D deep regression model (DenseNet-121) and the 3D segmentation model (nnU-Net) is provided in Supplementary table 1. This demonstrates the potential of our deep regression framework to predict clinical outcomes, such as OS, which are related to TMTV but cannot be directly derived from whole-body PET/CT images.

Automated age prediction from whole-body FDG-PET/CT projections is another difficult yet interesting task due to the complex relationship between age and underlying anatomical as well as metabolic changes. The presence of tumors can further complicate age prediction, as it often correlates with age. Given the complex inter-dependence between age and various clinical outcomes, accurately predicting age from FDG-PET/CT projections has the potential to provide deeper insights into patient prognosis[19–21]. Similar to TMTV, we observed that integrating additional

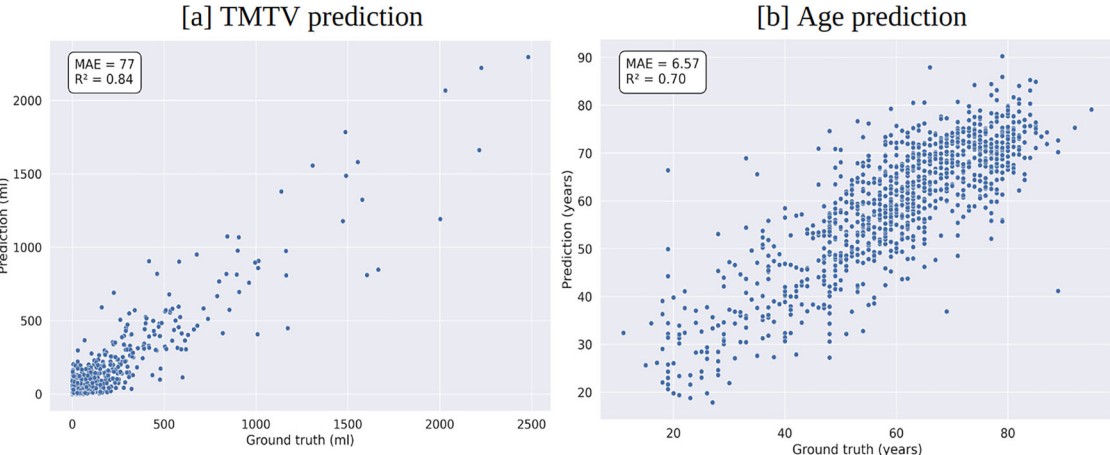

**Fig. 4 | Comparison of ground truth and predicted values from the best-performing model.** Scatterplots for (**a**) total metabolic tumor volume (TMTV) and (**b**) age prediction.

**Table 4 | Comparison of sex and diagnosis status classification results from 10-fold cross-validation using tissue-wise CT and SUV projections from various angles**

| Prediction | Inputs ($CT^{meanIP}$/$SUV^{MIP}$) | | | | | AUC | Sensitivity | Specificity | Precision |
|---|---|---|---|---|---|---|---|---|---|
| | original | bone | lean | adipose | air | | | | |
| | | | | | | (%) | (%) | (%) | (%) |
| [1] Sex | ✓/✓ | ✓/✓ | ✓/✓ | ✓/✓ | ✓/✓ | 1.00 (*) | 1.00 | 1.00 | 1.00 |
| [2] Sex | –/✓ | –/– | –/– | –/– | –/– | 0.99 | 0.98 | 0.99 | 0.99 |
| [3] Sex | ✓/– | –/– | –/– | –/– | –/– | 0.99 | 0.99 | 0.99 | 0.99 |
| [4] Sex | ✓/– | ✓/– | ✓/– | ✓/– | ✓/– | 0.99 | 0.98 | 1.00 | 1.00 |
| [5] Sex | –/✓ | –/✓ | –/✓ | –/✓ | –/✓ | 0.99 | 0.99 | 0.99 | 0.99 |
| [6] Diagnosis status | –/✓ | –/✓ | –/✓ | –/✓ | –/✓ | 0.95 (*) | 0.92 | 0.90 | 0.91 |
| [7] Diagnosis status | –/✓ | –/– | –/– | –/– | –/– | 0.91 | 0.79 | 0.90 | 0.89 |
| [8] Diagnosis status | ✓/– | ✓/– | ✓/– | ✓/– | ✓/– | 0.74 | 0.66 | 0.76 | 0.73 |

The inputs include combinations of original, bone, lean, adipose, and air channels, evaluated using DenseNet-121 model. DeLong's test was conducted between model 1 (proposed) and 3 (baseline) for sex and model 6 (proposed) and 7 (baseline) for diagnosis status classification, where '*' indicated statistical significance ($p$-value < 0.05).

channels improved the prediction of age from $R^2 = 0.42$ to 0.70 and r from 0.65 to 0.84 (Steiger's test, $p < 0.05$), as shown in Table 3. However, unlike TMTV, both CT and SUV projections were found useful. The reason was likely that CT offers detailed anatomical insights, while SUV provide metabolic information across the body, both of which are important to estimate the age. Our deep regression approach has shown its capability to predict age, which is not directly measurable from medical images. This highlights the method's potential to uncover latent information embedded within imaging data, thereby facilitating comprehensive patient assessments, including future potential of predicting patient outcomes.

The introduction of oblique projections, in addition to coronal and sagittal projections, improved age prediction by reducing the MAE from 6.57 to 6.41 and increasing the $R^2$ from 0.70 to 0.72. However, for TMTV prediction, the number of additional directions was not found to be important. To determine whether both sagittal and coronal projections were important for prediction of TMTV and age, or if a single projection could yield similar performance, we cross-validated a network using only the coronal projections ($\Theta = 0°$). This network exhibited lower performance in comparison to the prior trained with the combination of sagittal and coronal projections (see Tables 1 and 3). The performance drop was evident in both TMTV ($\Delta R^2 = 0.07$) and age ($\Delta R^2 = 0.03$) prediction. Finally, we also found that our network significantly outperformed the simple approach of guessing the mean of the entire cohort as assessment of the TMTV (MAE = 203 ml) and age (MAE = 44.64 years).

Expanding on TMTV prediction, our study also aimed to estimate total lesion count, which is associated with clinical outcomes[17]. The lesion count model achieved an $R^2$ of 0.90, which surpassed the performance of the 3D segmentation network (see Table 2) highlighting its potential for future applications in clinical outcome prediction. Additionally, we explored tissue-wise SUV and CT projection-based classification of sex and diagnosis status. The proposed model achieved an AUC of 1.00 for sex classification and 0.95 for diagnosis status classification, significantly outperforming baseline methods in AUC for both tasks (Delong's test, $p < 0.05$). Notably, the model achieved perfect performance in sex classification, highlighting its strong suitability for this dataset. These findings demonstrate the ability of deep regression models to extract tissue-specific lesion information, which could play a crucial role in risk prediction and assessment of therapy response.

Saliency analysis is essential for enabling interpretability of the deep regression model, as it allows us to identify the specific image regions that contribute to each prediction. This understanding increases our confidence that the highlighted areas represent meaningful features rather than random artifacts. In this study, we focused on TMTV, age, sex, and diagnosis status to ensure that the network highlights regions associated with these target variables before applying the approach to clinical outcomes. For TMTV prediction (see Supplementary Fig. 1) and diagnosis status classification (see Supplementary Fig. 4), the model primarily focused on tumor regions, which supported our hypothesis and aligned well with logical expectations.

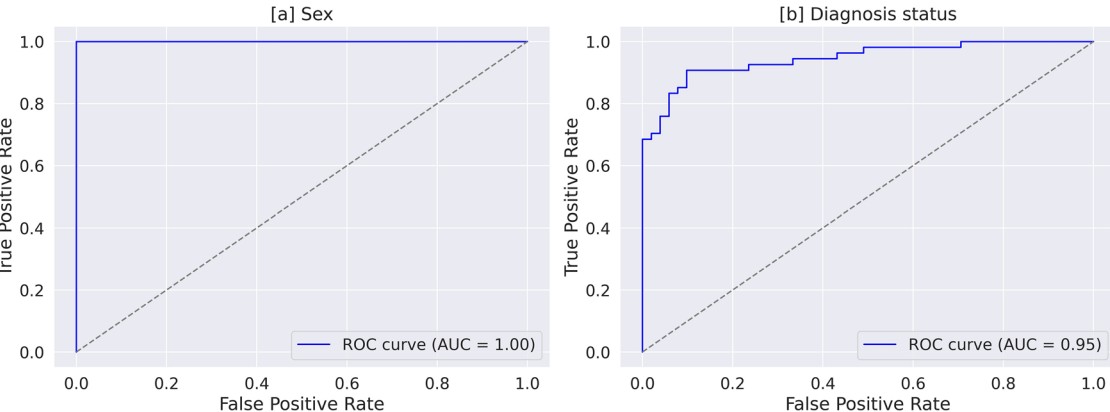

**Fig. 5 | Receiver operating characteristic (ROC) curves for classification tasks.** ROC curves illustrating the classification performance for (**a**) sex and (**b**) diagnosis status for one of the folds.

**Fig. 6 | Aggregated saliency maps highlighting regions contributing to model predictions.** Heatmaps illustrating aggregated saliency maps (cohort-level saliency analysis) along coronal and sagittal directions for (**a**) total metabolic tumor volume (TMTV), (**b**) age, (**c**) sex, and (**d**) diagnosis status. These maps were derived from the best-performing model and highlight recurring regions of interest contributing to predictions based on tissue-wise FDG-PET/CT projections.

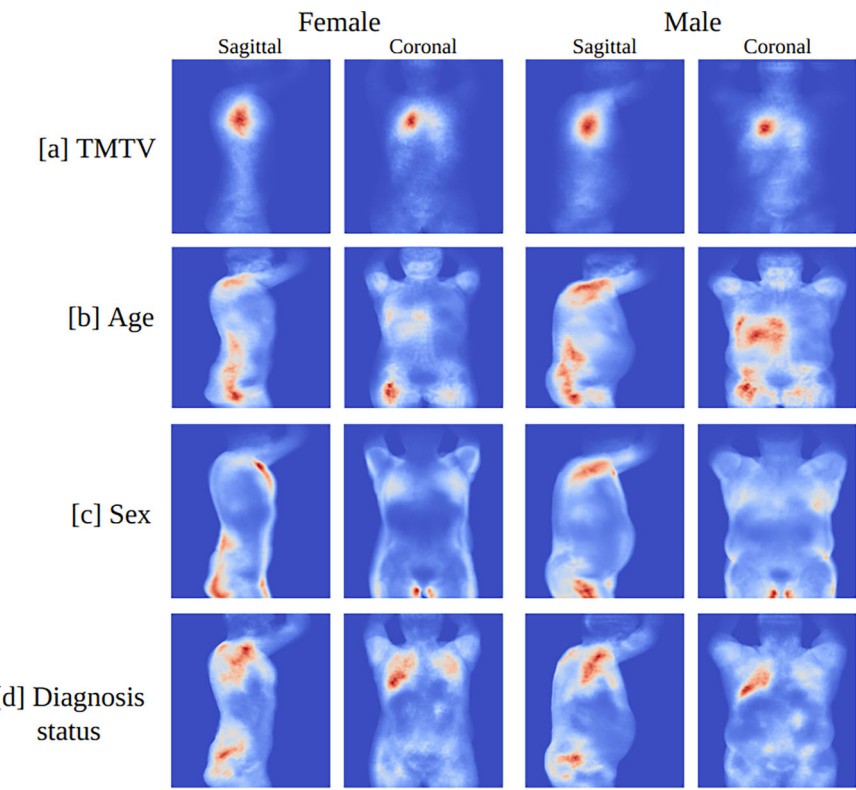

We also investigated the cases where the CNN largely mis-estimated the TMTV (see Supplementary Fig. 5) or diagnosis status (see Supplementary Fig. 7). We found that most of these cases contained either very small tumors with low FDG uptake or high FDG uptake regions that resembled tumors but were not. In the case of age prediction, the model primarily focused on the following regions: the liver, thigh muscles, areas near the femur head and pelvis, as well as the shoulder muscles (see Supplementary Fig. 2). Upon investigating instances where the CNN performed poorly (see Supplementary Fig. 6), we found that these cases typically involved issues with the input image, such as widespread tumors covering the entire body or unusual body structure, or a high tumor burden, which affected the network's ability to accurately predict age. Sex classification was the simplest task, and after analyzing the saliency maps, we found that the network primarily focused on the genitals and chest region for classification (see Supplementary Fig. 3). These findings were further validated through cohort saliency analysis, as shown in Figure 6, which revealed that the same regions were highlighted in

the entire cohort for the prediction tasks, consistent with the individual saliency maps.

The promising results from our proof-of-concept study suggest the potential of the deep regression framework in future prediction of clinical outcomes that are not readily derivable from medical images, such as OS. This is supported by the established associations between OS and clinical parameters such as TMTV, lesion count, age, and sex. This capability is crucial as OS serves as a critical endpoint in many clinical studies, and reliable prediction from imaging data has the potential to transform patient management and prognostication strategies. Future clinical implications of our study extend to improving predictive accuracy, enabling non-invasive assessment, monitoring treatment response, and facilitating risk stratification.

Implementing the proposed framework in real-world clinical settings requires diverse and extensive datasets for robust model training and validation. Integration into clinical practice necessitates collaboration with

radiologists for rigorous validation and adaptation to clinical workflows. Additionally, obtaining regulatory approval is essential to ensure the safety, efficacy, and compliance of the framework with healthcare standards. Overcoming these challenges involves iterative development, continuous validation against clinical standards, and ensuring scalability and generalizability across different patient populations and medical centers.

There are several limitations to our study. Firstly, the TMTV prediction network was trained exclusively on tumor-positive scans, potentially biasing the framework to predict non-zero TMTV values even in tumor-free cases. One potential solution is to use a diagnosis status classification network to filter out disease-free scans before regression; however, with an AUC of 0.95 as shown in Table 4, there are still some false positives and negatives, which is not ideal. Secondly, our projection-based technique is most suitable for identifying lesions with high FDG avidity. The use of a MIP can obscure low-intensity or small lesions that are readily apparent on 3D slice-by-slice analysis. While our multi-tissue projection approach mitigates this by preserving more information than a standard MIP, the fundamental limitation of projecting 3D data into a 2D space still remains. Thirdly, all data came from a single medical center, introducing potential confounding factors specific to the scanning protocol that could affect the network's generalizability. Lastly, the standardization of manually recorded TMTV values could vary due to differences in radiologists' annotations, particularly at tumor boundaries. This variability introduces errors since the network lacks limited standardization of TMTV reference for optimization, though the manual annotations are representative of typical ground truth data.

## Ethics approval and consent to participate

Ethical approval to conduct retrospective image analysis on the autoPET dataset was obtained from the Swedish Ethical Review Authority (Dnr 2023-02312-02). The study was conducted in accordance with relevant guidelines and regulations, including the Declaration of Helsinki.

## Conclusions

We have introduced a tissue-wise FDG-PET/CT projection-based approach for the prediction of parameters associated with clinical outcome, using a CNN. Compared to single projections, multi-channel projections improved the prediction of TMTV, age, sex, and diagnosis status. To introduce transparency and explainability, saliency analysis was conducted to visualize the image regions that influenced the CNN's predictions. The results indicated that the highlighted regions correspond to meaningful features aligned with our hypothesis, rather than seemingly random regions. This underscores our method's potential for explainable future predictions of clinical outcomes, such as OS, which could support informed decisions in clinical practice.

## Data availability

The image data that support the findings of this study are available in The Cancer Imaging Archive (TCIA) as part of the AutoPET dataset, with the identifier doi:10.7937/gkr0-xv29 (https://www.cancerimagingarchive.net/collection/fdg-pet-ct-lesions/)[9]. Source data for Figure 2 is in Supplementary Data 1. Source data for Figure 3 is in Supplementary Data 2. Source data for Figure 4 is in Supplementary Data 3 and 4. Source data for Figure 5 is in Supplementary Data 5 and 6.

## Code Availability

The code is available in the following GitHub repository: https://github.com/sambittarai/Tumor-segmentation-from-PET-CT-followed-by-clinical-parameter-estimation/tree/main/Clinical%20parameter%20estimation[36].

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

## Acknowledgements
This study was supported by the Swedish Cancer Society (23 3123Pj 01 H), Analytic Imaging Diagnostics Arena (AIDA), Lions Cancer Fund Uppsala and Makarna Eriksson foundation.

## Author contributions
Conceptualization: S.T. and J.K.; Method development: S.T. and J.K.; Image analysis: S.T., E.L., N.A., H.A., J.K.; Interpretation of results: S.T., E.L., N.A. and J.K.; Funding acquisition: H.A. and J.K.; Supervision: E.L., R.S., H.A., J.K.; Manuscript preparation: S.T.; Manuscript revision, reading, and approval of the final manuscript: all authors.

## Funding

## Competing interests
Håkan Ahlström and Joel Kullberg are affiliated with Antaros Medical AB, where they are employed part time or hold equity/stocks.
