## [Transparent Peer Review file · Communications Medicine]

Interpretable predictions from whole-body FDG-PET/CT using parameters associated with clinical outcome

Corresponding Author: Dr Sambit Tarai

Version 0:

Reviewer comments:

Reviewer #1

(Remarks to the Author)

Dear Editorial Office,

Thanks for considering me to review this paper.

The distinguished authors in “Interpretable deep regression-based prediction of parameters associated with clinical outcome from tissue-wise FDG-PET/CT projections” tried to evaluate their novel tissue-wise approach to predict clinical outcomes. This is in continuation of the previously published, well-done work by their research group on metabolic volumetry.

My fundamental concern is the way this paper is structured (to make it short, being clinical instead of technical). The authors started with the limitations of conventional methods, such as Cox regression, in outcome prediction. Cox regression is for the “overtime” prediction of the events and is a prognostication method. However, predicting patients’ disease status, TMTV, age and sex does not predict the outcome of patients in future as is about the current status/characteristics of the individuals. Furthermore, although these parameters have documented roles in patient prognostication, predicting them themselves is not prognostication on its own. To make it simple, I cannot say direct clinical inspection to classify the sex of a patient is a prognostic tool for predicting the patient’s overall survival at the time of lymphoma diagnosis.

Other points are:

- Introduction: Again, this should be totally revised, as mentioning the limitations of Cox regression at the beginning is completely misleading. The cited reference (#1) proposed neural network-based “time-to-event” prediction, which is a valid thing to be compared to the Cox regression.
- Introduction: Age and sex are mandatory things when a physician reads an image and are ALWAYS provided as part of patient registration. However, I assume you may want to talk about a non-labelled database out there that provided the full sets of FDG PET/CT images with their diagnoses and suddenly dropped the age (which is, honestly, rare). But even this scenario would not be the case for sex as it is “visibly present”. The only clinically beneficial parameter would be “disease vs non-disease”, which is actually the same as the TMTV calculation. If you can predict TMTV, you can say disease vs non-disease without any additional info (you can tell it over the phone, and the clinician can say the status!). Thus, overall, the only valuable parameter to predict is TMTV, which is already presented in the other published paper. You can draft this paper as an enhanced version of the previously introduced tissue-wise method.
- Methods: Relevant, but following the discussed limitations of the objectives.
- Results: Follow the methods.
- Discussion: Same issues as discussed for the introduction. “Sex classification was the simplest task, and after analyzing the saliency maps, we found that the network primarily focused on the genitals and chest region for classification”; Well, I cannot agree more!
- Discussion: “Given the known association between TMTV, lesion count, age, and sex with OS, our approach could enhance the accuracy of OS predictions.”; Again, although this is correct in a way, it is not the best scientific way to bold the value of the findings. I want to re-mention my example that a clinician cannot publish a paper by saying, “given the known associations between sex and cancer prognosis, my direct clinical inspection can enhance the accuracy of OS predictions”.

Here are my suggestions for the distinguished authors:

- First and foremost, make this a technical paper focusing on image labelling and patient classification rather than presenting it as a prognostication tool.
- If you could significantly improve your previously published tissue-wise model, publish it as a paper indicating this main fact.
- Focus on the things that you found are related to ageing and are not in the current literature. That may open up new fields for clinicians working on ageing and can be of significant interest.
- Avoid focusing on the anatomical differences between the two sexes, as they are well-known.

Reviewer #2

(Remarks to the Author)

In this study, the authors evaluated tissue-wise FDG PET-CT projections to predict several features of interest (age, sex, diagnostic status, TMTV). They developed an original approach, feeding their deep neural network with multi-channel SUV / CT 2D projections (tissue-wise projections of air, bone, lean and adipose tissue). The framework trained and validated on the auto-PET dataset (1014 PET/CT series) from the Cancer Imaging Archive, provided very promising results. Furthermore, the saliency maps provided with the Grad-CAM method highlighted the consistence of the network in making its features' predictions.

The study is well written, well designed, and is of interest in the field of AI applied to PET imaging.

Before potential publication, several methodological points should be clarified, or discussed:

1) Patients' cohort: here we have three patients' cohorts: lymphoma, lung cancer and melanoma patients. These populations typically show, from a PET perspective, particular patterns, which may not be very challenging to classify on MIPs over normal ones (respectively lymphoproliferative uptakes more or less disseminated, lung nodules, and small foci –metastases in transit). Furthermore, in practice, entire body PET is typically performed for melanoma, not for lymphoma and NSCLC (torso PET from vertex to mid-tigh). In this context,

Can the authors demonstrate that accurate classification of the disease status is based on features rather than a mix between features and FOV of the MIP?

What would be the performance with more subtle MIPs oncological diseases (digestive or gynaecologic with peritoneal carcinomatosis, which show sometimes very subtle FDG PET uptakes, mostly seen on CT part or 3D PET slices.

The classification task was binary (cancer positive or negative): would the framework be able to discriminate sub entities only based on MIP?

2) How to deal with CT abnormalities without significant FDG avidity (like small lesions or particular histological subtypes)?

3) Controls cohort: the control cohort included negative MIPs. In practice, numerous conditions may mimic cancer and represent challenging classification tasks for neural networks: granulomatosis, infectious diseases etc... What would be the performance of the framework in real-life conditions (no caricatured cases confronting very abnormal MIP versus completely normal MIPs)? It is of importance if we want to deploy such a process in future clinical practice, not only for OS studies with very "pure" and very highly selected patient's cohorts.

4) The monocentric design: has mentioned by the authors, only one PET hardware was used in this study. Consequently, the robustness of the framework to the hardware or image post-processing is unknown.

5) It could have been of interest to develop foundation model with more various downstream tasks (multi-tissue labelling, multi-organ or lesions segmentation tasks, more PET features extraction...). For your interest: Pai, S., Bontempi, D., Hadzic, I. et al. Foundation model for cancer imaging biomarkers. Nat Mach Intell 6, 354–367 (2024). <https://doi.org/10.1038/s42256-024-00807-9>.

Reviewer #3

(Remarks to the Author)

This study introduced a tissue-wise FDG-PET/CT approach that effectively automated the prediction of clinical outcome-related parameters and incorporated saliency analysis for model interpretability. The manuscript employs tissue-wise multi-channel projections to implement both regression and classification predictions. By comparing various CNN models and demonstrating advantages over existing multi-channel projection approaches in the literature, this study shows significant strengths in experimental design, methodological implementation, and clinical relevance. It provides valuable insights for the nuclear medicine field. The writing quality and logical flow have reached a publishable standard.

Version 1:

Reviewer comments:

Reviewer #2

(Remarks to the Author)

I thank the authors for their clear responses to the various points I initially raised, which are satisfactory to me. I have no further comments to add

Reply to Reviewer comments

Reviewer 1

My fundamental concern is the way this paper is structured (to make it short, being clinical instead of technical). The authors started with the limitations of conventional methods, such as Cox regression, in outcome prediction. Cox regression is for the “overtime” prediction of the events and is a prognostication method. However, predicting patients’ disease status, TMTV, age and sex does not predict the outcome of patients in future as is about the current status/characteristics of the individuals. Furthermore, although these parameters have documented roles in patient prognostication, predicting them themselves is not prognostication on its own. To make it simple, I cannot say direct clinical inspection to classify the sex of a patient is a prognostic tool for predicting the patient’s overall survival at the time of lymphoma diagnosis.

Answer:

We would like to thank Reviewer 1 for their thoughtful comments and for highlighting areas where the intent of our manuscript was not fully clear. We appreciate the opportunity to clarify our study's purpose and have revised the manuscript accordingly.

We believe there might be a misunderstanding regarding the primary objective of our work, for which we take responsibility due to a lack of clarity in our original introduction. Therefore we have now revised the introduction completely. Our study was not designed to create a new prognostic model for patient survival. Instead, it was a proof-of-concept study showing that our deep regression methodology can predict key clinical characteristics of a patient (such as TMTV, age, sex, disease status) directly from medical images.

We completely agree with the reviewer that predicting a patient's sex or TMTV is not, in itself, a prognostication of their overall survival. However, our goal was to show that our model can automatically extract clinically relevant features, which are used by clinicians for prognostication. We have now restructured the manuscript to explicitly state that this was a proof-of-concept study for feature prediction, not a survival prognostication study. We have also removed any phrasing that could be misconstrued as claiming direct prognostication.

We have addressed the specific concerns raised by the reviewer in the responses provided below. We kindly ask the reviewer to review these answers for a clearer understanding.

Other points are:

[1] Introduction: Again, this should be totally revised, as mentioning the limitations of Cox regression at the beginning is completely misleading. The cited reference (#1) proposed neural network-based “time-to-event” prediction, which is a valid thing to be compared to the Cox regression.

Answer:

- We have now completely revised the introduction section as per the reviewer’s request.
- We have now also removed the mentioning of the limitations of the cox regression at the beginning of the introduction.

[2] Introduction: Age and sex are mandatory things when a physician reads an image and are ALWAYS provided as part of patient registration. However, I assume you may want to talk about a non-labelled database out there that provided the full sets of FDG PET/CT images with their diagnoses and suddenly dropped the age (which is, honestly, rare). But even this scenario would not be the case for sex as it is “visibly present”. The only clinically beneficial parameter would be “disease vs non-disease”, which is actually the same as the TMTV calculation. If you can predict TMTV, you can say disease vs non-disease without any additional info (you can tell it over the phone, and the clinician can say the status!). Thus, overall, the only valuable parameter to predict is TMTV, which is already presented in the other published paper. You can draft this paper as an enhanced version of the previously introduced tissue-wise method.

Answer:

We agree with the reviewer that age and sex are routinely available in clinical practice. However, the primary aim of this study was not to develop a tool for predicting these specific parameters. Instead, this was a proof-of-concept methodological study. The value of predicting parameters that are known to be present in the image data such as sex, TMTV, lesion count, and disease status, is twofold:

- It allows us to quantify how accurately the proposed methodology can predict these clinically relevant variables.
- It enables us to evaluate the performance of our novel interpretability approach, cohort saliency analysis, by applying it to tasks where we have prior knowledge of the key image regions contributing to the predictions.

The central contribution of this work is the validation of a novel tissue-wise 2D projection framework as an efficient and interpretable alternative to processing computationally intensive 3D volumes. The prediction of diverse parameters such as TMTV, lesion count, age, and sex serves a specific methodological purpose, for e.g.:

- The ability to predict age and sex from images acts as a robust internal validation for our model. Since these attributes are not the primary focus of a PET/CT scan for oncology, a model that accurately predicts them could learn subtle correlations within the imaging data. This suggests that the framework is capable of learning the complex patterns necessary for future predictions of more challenging clinical outcomes (e.g., overall survival, treatment response) that are related to these clinical parameters and are not directly visible from the images.
- While our previous paper introduced the tissue-wise concept for TMTV prediction, this manuscript significantly expands the scope. It is not only an enhancement but a comprehensive validation of the method's broader potential. By showing its efficacy across multiple prediction tasks and incorporating extensive cohort-level saliency analysis for interpretability.
- In summary, predicting age, sex, TMTV, lesion count serves as a proof of concept towards developing models capable of forecasting clinical outcomes, particularly in cases where imaging correlates are unknown or highly complex.

We have now taken the following actions to address the reviewer's concern:

- For the ease of understanding we have now modified the title of the manuscript.
- The below highlighted paragraph in the introduction section addresses the importance of predicting age and sex:
 - “Studies have shown that age and sex are frequently incorporated as covariates in Cox proportional hazards models for survival analysis. \cite{deo2021survival} \cite{cuccaro2014prognostic} \cite{tas2013age}. Unlike TMTV, which is visibly present in the image, age prediction is relatively challenging as it is not directly derivable from the images. If the network successfully can predict age, this indicates that it has learned associations that may not be obvious to humans. For instance, the discrepancy between predicted age and chronological age could offer information regarding the presence and severity of the disease. While the ability to predict TMTV and age demonstrates the capability of the deep regression framework, it also highlights the network's potential to capture meaningful associations between imaging data and clinical outcomes. Similarly classifying patients based on their sex and diagnosis status (disease vs non-disease) is also crucial. It not only serves as a foundation but also provides the opportunity for more complex categorizations in the future, including positive vs negative treatment outcomes.”

[3] Methods: Relevant, but following the discussed limitations of the objectives.

Answer: Thank you so much. We hope that our updates and clarification of other sections will make the method section more in line with the study aims.

[4] Results: Follow the methods.

Answer: Thank you so much. We hope that our updates and clarification of other sections will make the result section more in line with the study aims and methods.

[5] Discussion: Same issues as discussed for the introduction. “Sex classification was the simplest task, and after analyzing the saliency maps, we found that the network primarily focused on the genitals and chest region for classification”; Well, I cannot agree more!

Answer: As outlined in the response to question 2, this was a proof-of-concept study in which we developed a deep regression-based framework to predict parameters associated with clinical outcomes. For further details, please refer to the answer provided in question 2.

We hope that our updates and clarification of other sections will make the discussion section more in line with the study aims.

[6] Discussion: “Given the known association between TMTV, lesion count, age, and sex with OS, our approach could enhance the accuracy of OS predictions.”; Again, although this is correct in a way, it is not the best scientific way to bold the value of the findings. I want to re-mention my example that a clinician cannot publish a paper by saying, “given the known associations between sex and cancer prognosis, my direct clinical inspection can enhance the accuracy of OS predictions”.

Answer:

We have removed the sentence highlighted by the reviewer and revised the paragraph accordingly, as shown below.

“The promising results from our proof-of-concept study suggest the potential of the deep regression framework in future prediction of clinical outcomes that are not readily derivable from medical images, such as OS. This is supported by the established associations between OS and clinical parameters such as TMTV, lesion count, age, and sex.”

Suggestions for the distinguished authors:

[7] First and foremost, make this a technical paper focusing on image labelling and patient classification rather than presenting it as a prognostication tool.

Answer: We now hope and believe that it is even clearer that this manuscript represents a technical proof-of-concept study.

[8] If you could significantly improve your previously published tissue-wise model, publish it as a paper indicating this main fact.

- Focus on the things that you found are related to ageing and are not in the current literature. That may open up new fields for clinicians working on ageing and can be of significant interest.
 - What did we find about aging?
 - Mention some things from the saliency analysis.
 - Liver volume is associated with age (UKBB), we confirm from cohort saliency maps.

Answer: This is indeed a significant extension of previous work. (See answers above).

We would like to clarify that this is not an aging study. The primary purpose of the age prediction task was to assess how effectively our methodology could estimate age and whether the corresponding cohort saliency analysis could identify image regions reasonably associated with age. It is important to note, however, that “normal” aging patterns are difficult to define in patients affected by cancer and potentially influenced by treatment, making interpretation more complex in this context.

[9] Avoid focusing on the anatomical differences between the two sexes, as they are well-known.

Answer: We hope that our updated text now makes it much more clear why the performance of the sex prediction and its saliency is so relevant for this study.

Reviewer 2

In this study, the authors evaluated tissue-wise FDG PET-CT projections to predict several features of interest (age, sex, diagnostic status, TMTV). They developed an original approach, feeding their deep neural network with multi-channel SUV / CT 2D projections (tissue-wise projections of air, bone, lean and adipose tissue). The framework trained and validated on the auto-PET dataset (1014 PET/CT series) from the Cancer Imaging Archive, provided very promising results. Furthermore, the saliency maps provided with the Grad-CAM method highlighted the consistence of the network in making its features' predictions.

The study is well written, well designed, and is of interest in the field of AI applied to PET imaging.

Answer: We would like to thank Reviewer 2 for such a nice summary of our study and their thoughtful comments. We have addressed these points below and have revised the manuscript accordingly.

Before potential publication, several methodological points should be clarified, or discussed:

[1] Patients' cohort: here we have three patients' cohorts: lymphoma, lung cancer and melanoma patients. These populations typically show, from a PET perspective, particular patterns, which may not be very challenging to classify on MIPs over normal ones (respectively lymphoproliferative uptakes more or less disseminated, lung nodules, and small foci –metastases in transit). Furthermore, in practice, entire body PET is typically performed for melanoma, not for lymphoma and NSCLC (torso PET from vertex to mid-tigh). In this context,

[a] Can the authors demonstrate that accurate classification of the disease status is based on features rather than a mix between features and FOV of the MIP?

Answer:

This is an excellent point, and we thank the reviewer for raising the critical issue of "shortcut learning," where a model might look at confounding variables like the Field-of-View (FOV) rather than genuine pathological features.

While it is difficult to completely exclude this possibility, we have taken several steps to investigate and mitigate it:

- Our "negative" (disease-free) cohort contained images from all three cancer types (lymphoma, lung cancer, melanoma) acquired with their respective standard FOVs. Therefore, a simple FOV-based classification (e.g., "head-to-toe = positive, torso = negative") would not be effective, as both positive and negative examples exist for each FOV type.
- Our qualitative analysis of saliency maps indicates that the model's attention is consistently focused on regions with physiologically plausible or pathological uptake rather than on the image boundaries that define the FOV.

[b] What would be the performance with more subtle MIPs oncological diseases (digestive or gynaecologic with peritoneal carcinomatosis, which show sometimes very subtle FDG PET uptakes, mostly seen on CT part or 3D PET slices).

Answer:

- Our study focused on cancer types that typically present with FDG-avid and structurally discernible patterns on MIPs. Our model's performance on diseases with more subtle or diffuse uptake, such as peritoneal carcinomatosis, remains unknown and is out of scope of this study.
- Even though this has not been evaluated here, it is likely that subtle FDG PET uptakes and changes only found in CT would suffer more from the multi-tissue projection technique utilized in this study.

[c] The classification task was binary (cancer positive or negative): would the framework be able to discriminate sub entities only based on MIP?

Answer:

- While this was not the primary aim of our current proof-of-concept study, we believe it is a promising direction. It is well-established that these three cancer types often have distinct patterns of dissemination on PET. A previous study by Hanna Jönsson et al. (<https://pubmed.ncbi.nlm.nih.gov/38007471/>) demonstrated that these different cancer types show different patterns on a cohort level.
- Although we have not trained or validated a multi-class model in this work, the features learned by our model for the binary "positive/negative" task likely capture these underlying disease-specific patterns. This could be an exciting avenue for future research.

[2] How to deal with CT abnormalities without significant FDG avidity (like small lesions or particular histological subtypes)?

Answer: The current projection technique is not well-suited for CT abnormalities that lack significant FDG avidity. These cases fall outside the scope of our method, and we have not specifically tested its performance on such CT-only findings as it was out of scope for this study.

We have now added the following sentence to the discussion section (as limitations):

“Secondly, our projection-based technique is most suitable for identifying lesions with high FDG avidity. The use of a MIP can obscure low-intensity or small lesions that are readily apparent on 3D slice-by-slice analysis. While our multi-tissue projection approach mitigates this by preserving more information than a standard MIP, the fundamental limitation of projecting 3D data into a 2D space still remains.”

[3] Controls cohort: the control cohort included negative MIPs. In practice, numerous conditions may mimic cancer and represent challenging classification tasks for neural networks: granulomatosis, infectious diseases etc... What would be the performance of the framework in real-life conditions (no caricatured cases confronting very abnormal MIP versus completely normal MIPs)? It is of importance if we want to deploy such a process in future clinical practice, not only for OS studies with very “pure” and very highly selected patient’s cohorts.

Answer: We acknowledge the reviewer’s concern regarding the representativeness of the control cohort. While the dataset includes a large number of control scans (approximately 500), these individuals are not healthy volunteers, but we have limited information about their clinical status. We assume they underwent imaging for some clinical indication (or as a post treatment follow-up examination), which introduces variability in a similar way as expected in a clinical setting. A small subset of cases includes imaging artifacts, but their number is insufficient to draw meaningful conclusions. To fully assess the framework’s performance under real-world conditions, further validation on a more diverse cohort with known health status is necessary, which was out of scope of the current study.

[4] The monocentric design: has mentioned by the authors, only one PET hardware was used in this study. Consequently, the robustness of the framework to the hardware or image post-processing is unknown.

Answer: We acknowledge that the monocentric design of this study is one of the limitations of this work. However, this constraint was beyond our control, as we did not have access to datasets from other institutions, despite our intention to include more diverse sources. We have also clearly mentioned this as a limitation of our study in the manuscript.

[5] It could have been of interest to develop foundation model with more various downstream tasks (multi-tissue labelling, multi-organ or lesions segmentation tasks, more PET features extraction...). For your interest: Pai, S., Bontempi, D., Hadzic, I. et al. Foundation model for cancer imaging biomarkers. Nat Mach Intell 6, 354–367 (2024). <https://doi.org/10.1038/s42256-024-00807-9>).

Answer: Thank you for this valuable suggestion. Developing a foundation model for a wider range of downstream tasks is an excellent idea and aligns with our long-term goal. However, expanding the scope in this manner was beyond the resources and objectives of the current study, and we have therefore designated it for future work.

Reviewer 3

This study introduced a tissue-wise FDG-PET/CT approach that effectively automated the prediction of clinical outcome-related parameters and incorporated saliency analysis for model interpretability. The manuscript employs tissue-wise multi-channel projections to implement both regression and classification predictions. By comparing various CNN models and demonstrating advantages over existing multi-channel projection approaches in the literature, this study shows significant strengths in experimental design, methodological implementation, and clinical relevance. It provides valuable insights for the nuclear medicine field. The writing quality and logical flow have reached a publishable standard.

Conclusion: Acceptance after minor revisions

Answer: We would like to thank Reviewer 3 for such a nice summary of our study and their thoughtful comments. We have addressed these points below and have revised the manuscript accordingly.

Overall Assessment: This study introduced a tissue-wise FDG-PET/CT approach that effectively automated the prediction of clinical outcome-related parameters and incorporated saliency analysis for model interpretability. The manuscript employs tissue-wise multi-channel projections to implement both regression and classification predictions. By comparing various CNN models and demonstrating advantages over existing multi-channel projection approaches in the literature, this study shows significant strengths in experimental design, methodological implementation, and clinical relevance. It provides valuable insights for the nuclear medicine field. The writing quality and logical flow have reached a publishable standard. The authors are requested to address the following minor revisions prior to final acceptance:

[1] In line 47, the notation "¹⁸F-FDG" should be revised to use superscript formatting for the isotopic number ¹⁸F-FDG. This correction should be applied consistently throughout the manuscript.

Answer: Thank you so much for pointing out. We have now updated it to use the correct notation.

[2] In Section 2.3, is SUV mapped based on CT tissue segmentation? Please explicitly describe the spatial correspondence mechanism between CT-derived tissue segmentation and SUV quantification.

Answer:

- Yes, SUV (Standardized Uptake Value) is mapped based on CT-derived tissue segmentation. The spatial correspondence between CT segmentation and SUV quantification is established through the inherent co-registration of PET and CT images, which are typically acquired simultaneously on hybrid PET/CT scanners. This ensures that both modalities are spatially aligned, allowing voxel-wise correspondence across the two datasets.
- CT images are used to segment anatomical structures by applying Hounsfield Unit (HU) thresholding. This segmentation yields binary or probabilistic masks for different tissue types (e.g., adipose tissue, muscle, bone). These masks are then applied to the co-registered PET image to extract SUV values specific to each tissue type.

[3] In line 193, The term "histogram distribution" should be replaced by "scatter plot".

Answer: Thank you for noting. We have now modified it in the manuscript.

[4] In Section 2.5.2, "several Networks" is used to evaluate TMTV and age prediction tasks to study the effectiveness of each channel. Different channels should also be valuable in the number of lesions, gender, and diagnostic status. Why did the authors not consider this? It can still be seen from Table 4 that the addition of CT helps to distinguish gender and diagnosis.

Answer: We agree with the reviewer that different input channels are likely to be valuable for predicting lesion count, sex, and diagnostic status. In this study, we conducted a limited number of ablation experiments for these specific tasks. While a more detailed channel-wise evaluation, similar to what we have performed for TMTV and age, could be valuable, we observed consistent trends indicating the usefulness of multi-channel inputs in those tasks. Based on this, and considering the time-intensive nature of such experiments, we chose not to extend the full ablation analysis to the remaining prediction tasks except TMTV and age.

[5] In Chapter 2.7, for TMTV and diagnostic state prediction tasks, only SUV projections at all organizational levels are included in the proposed model. In the conclusion part, CT data is actually used, so it is suggested to complete the description here.

Answer: We acknowledge that there is a confusion in section 2.7. We have now modified the text as shown below to clarify this further:

“In all the prediction tasks, the baseline model utilized either the SUV_{orig}^{MIP} or CT_{orig}^{MIP} as input, while the proposed model incorporated all tissue-wise CT and SUV projections.”

[6] In Figure 2, all TMTVS are divided into 10 different intervals with varying lengths. Please explain the reason for such partitioning.

Answer: Although TMTV is a continuous variable, we chose to divide it into 10 intervals of varying lengths to reflect the non-uniform distribution of tumor volumes in the dataset. This strategy ensures that each interval captures a meaningful range while maintaining a balanced representation of samples across categories. The objective is not to predict the exact TMTV value, but to estimate a value that closely approximates the true volume. This qualitative approach enables us to assess whether the model generalizes effectively and captures the inherent variability and noise present in real-world medical imaging.

[7] In the discussion, the MAE of TMTV under the current optimal model is 77ml. Please refer to more references to evaluate the importance of 77ml for clinical evaluation.

Answer: We acknowledge that the clinical relevance of a 77 ml MAE cannot be interpreted in isolation but must be evaluated in relation to the typical TMTV observed in the target patient population. Therefore, it is difficult to draw definitive conclusions without considering the distribution and clinical thresholds of TMTV in specific disease contexts. For instance, in patients with high tumor burden, a 77 ml deviation may be less impactful, whereas in cases with low TMTV, it could represent a significant proportion of the total volume. Since TMTV varies significantly across different cancer types and study cohorts, it is difficult to establish a universal benchmark for the clinical importance of a 77 ml error.

[8] Please standardize all reference entries according to the target journal's style guide.

Answer: We have now standardized all the references according to the journal's style guide.